# A Wolf in Another Wolf’s Clothing: Post-Genomic Regulation Dictates Venom Profiles of Medically-Important Cryptic Kraits in India

**DOI:** 10.3390/toxins13010069

**Published:** 2021-01-19

**Authors:** Kartik Sunagar, Suyog Khochare, R. R. Senji Laxme, Saurabh Attarde, Paulomi Dam, Vivek Suranse, Anil Khaire, Gerard Martin, Ashok Captain

**Affiliations:** 1Evolutionary Venomics Lab, Centre for Ecological Sciences, Indian Institute of Science, Bangalore 560012, Karnataka, India; suyogk@iisc.ac.in (S.K.); senjir@iisc.ac.in (R.R.S.L.); asaurabh@iisc.ac.in (S.A.); paulomidam@iisc.ac.in (P.D.); viveksuranse@iisc.ac.in (V.S.); 2Indian Herpetological Society, 7/47, Pune Satara Road, Pune 411009, Maharashtra, India; anilkhaire.ihs@gmail.com; 3The Liana Trust, Survey #1418/1419 Rathnapuri, Hunsur 571189, Karnataka, India; gerry@gerrymartin.in; 43/1 Boat Club Road, Pune 411001, Maharashtra, India; ashokcaptain@hotmail.com

**Keywords:** venom evolution, new krait species from India, antivenom therapy, venom proteomics, venom gland transcriptomics, Romulus’ krait

## Abstract

The Common Krait (*Bungarus caeruleus*) shares a distribution range with many other ‘phenotypically-similar’ kraits across the Indian subcontinent. Despite several reports of fatal envenomings by other *Bungarus* species, commercial Indian antivenoms are only manufactured against *B. caeruleus*. It is, therefore, imperative to understand the distribution of genetically distinct lineages of kraits, the compositional differences in their venoms, and the consequent impact of venom variation on the (pre)clinical effectiveness of antivenom therapy. To address this knowledge gap, we conducted phylogenetic and comparative venomics investigations of kraits in Southern and Western India. Phylogenetic reconstructions using mitochondrial markers revealed a new species of krait, Romulus’ krait (*Bungarus romulusi* sp. nov.), in Southern India. Additionally, we found that kraits with 17 mid-body dorsal scale rows in Western India do not represent a subspecies of the Sind Krait (*B. sindanus walli*) as previously believed, but are genetically very similar to *B. sindanus* in Pakistan. Furthermore, venom proteomics and comparative transcriptomics revealed completely contrasting venom profiles. While the venom gland transcriptomes of all three species were highly similar, venom proteomes and toxicity profiles differed significantly, suggesting the prominent role of post-genomic regulatory mechanisms in shaping the venoms of these cryptic kraits. In vitro venom recognition and in vivo neutralisation experiments revealed a strong negative impact of venom variability on the preclinical performance of commercial antivenoms. While the venom of *B. caeruleus* was neutralised as per the manufacturer’s claim, performance against the venoms of *B. sindanus* and *B. romulusi* was poor, highlighting the need for regionally-effective antivenoms in India.

## 1. Introduction

The Common Krait (*Bungarus caeruleus*), one of the ‘big four’ medically most-important Indian snakes, is well-known for causing numerous fatal envenomings in the country [1,2]. Bites from this nocturnal snake result in neuromuscular paralysis, which is primarily caused by the presence of β-bungarotoxin in the venom [3]. Considering the near country-wide distribution of this clinically important snake, *B. caeruleus* venoms are used for the manufacture of commercial Indian polyvalent antivenoms. Several reports of fatal envenomings by other superficially similar *Bungarus* species, which share a distribution range with *B. caeruleus*, have also come to light [4,5]. However, their venoms are not used for the manufacture of the life-saving antivenom, and the ‘big four’ antivenom is routinely used for the treatment of envenomings from such neglected species.

The Sind Krait (*Bungarus sindanus*) with 17 or 19 dorsal scale rows (DSR) at the mid-body, largely shares its geographical distribution with the phenotypically similar *B. caeruleus* (15 DSR) in Southeastern Pakistan and Western India (Rajasthan, Gujarat and Maharashtra). A subspecies of the Sind Krait—Wall’s Sind Krait (*B. sindanus walli*)—has also been described in the Gangetic Plains of Southeast Asia [6]. Although there have been several anecdotal reports of this subspecies in Western India [7,8,9] (also considered to be a distinct species, *Bungarus walli*, by some authors [10,11]), its phylogenetic identity remains to be validated.

In this study, by reconstructing species’ phylogenetic histories using mitochondrial markers (NADH-ubiquinone oxidoreductase chain 4 (ND4) and cytochrome *b* (cyt *b*)), we show that kraits with 17 DSR in Western India do not represent the subspecies *B. sindanus walli* as previously believed, but are genetically indistinguishable from the Sind krait (*B. sindanus*) in Pakistan. Additionally, our phylogenetic analyses recovered a new species of krait from Southern India, which we name Romulus’ krait (*Bungarus romulusi* sp. nov.). Through the use of comparative venom proteomics and venom gland transcriptomics, we show that the significant compositional differences in the venoms of cryptic kraits in Southern and Western India likely result from post-genomic regulatory mechanisms. Further, with the help of in vivo experiments in the murine model, we show that the venoms of *B. sindanus* and *B. romulusi* are amongst the most potently toxic snake venoms in the country, being over 11 and 6 times more potent than that of *B. caeruleus*, respectively. Consistent with previous clinical findings [4], our in vitro venom recognition assays and in vivo venom neutralisation experiments highlight the extremely poor efficacies of commercial antivenoms in treating *B. sindanus* envenomings. Thus, we highlight the importance of molecular phylogenetics in identifying clinically-important cryptic snake species and the pressing need for the development of regionally-effective antivenoms in India to counter the dramatic inter and intraspecific venom variations.

## 2. Results

To unravel the phylogenetic relationships and venom variation in cryptic kraits from Southern and Western India, we sampled scales, venoms, venom glands and physiological tissues from kraits with either 15- or 17-mid-body DSRs (Figure 1; Appendix A).

### 2.1. Phylogenetic Reconstructions

Phylogenetic reconstructions of two mitochondrial markers provided fascinating insights into the evolution of kraits in the Indian subcontinent (Figure 2 and Appendix A). The overall topology of *Bungarus* phylogeny was in complete agreement with the previously reported multilocus species tree [12]. Consistent with the literature, *B. sindanus* was recovered as a sister lineage to *B. caeruleus* (Bayesian Posterior Probability (BPP): 1; bootstrap (BS): 95). Interestingly, sequences from the 17-mid-body scale row krait from Maharashtra were found in the same clade as *B. sindanus* from Pakistan (BPP: 1; BS: 100). Surprisingly, however, *B. caeruleus* was found to be polyphyletic with four distinct clades (Figure 2 and Appendix A). The 15 DSR krait from Karnataka was found in a distinct clade to its counterparts from West Bengal and Maharashtra (BPP: 1; BS: 100). In contrast, the 15 DSR krait from Maharashtra was recovered as a sister lineage to *B. caeruleus* from Pakistan (BPP: 0.85; BS: 66). This clearly suggests that the 15 DSR krait from Karnataka is likely to be a genetically distinct species.

The estimation of evolutionary divergence between the mitochondrial sequences (ND4 and cyt *b*) strongly supported the aforementioned findings (Appendix A). Minor pairwise differences were noted between ND4 and cyt *b* sequences of the 17 DSR krait from Maharashtra and the *B. sindanus* from Pakistan (0.32% and 3.03%, respectively). This clearly shows that kraits with 17 DSR at mid-body in Western India are very closely related to *B. sindanus*, and assigning them to a subspecies (*B. sindanus walli*) or distinct species (*B. walli*) is questionable. Similarly, very few pairwise differences in nucleotides were documented between the 15 DSR krait from Maharashtra and its *B. caeruleus* counterparts from West Bengal (ND4: 2.69% to 3.25%). While the divergence between the ND4 and some of the cyt *b* sequences of the 15 DSR krait from Maharashtra and *B. caeruleus* from Pakistan were minimal (ND4: 2.35% to 3.73% and cyt *b*: 3.69% to 3.93%), cyt *b* sequence divergence between certain individuals were considerably large (5.61% to 6.55%). In complete contrast, significant differences were observed in the mitochondrial sequences of the 15 DSR krait from Karnataka, in comparison to its counterparts in Maharashtra (ND4: 8.6% and cyt *b*: 14.2%) and West Bengal (ND4: 6.96% to 7.09%). Considerable differences were also observed when these sequences from the Karnataka specimen with 15 DSR were compared with *B. caeruleus* from Pakistan (ND4: 7.72% to 8.30% and cyt *b*: 6.67% to 12.04%). This conclusively shows that the 15 DSR krait from Karnataka represents an unrecognised species. For his remarkable contribution to the field of herpetology, we name this new krait species as Romulus’ krait (*Bungarus romulusi* sp. nov.) in honour of Indian herpetologist, Romulus Whitaker.

### 2.2. Venom Proteomics

SDS-PAGE profiling of *B. sindanus* (Rajasthan and Maharashtra), *B. caeruleus* and *B. romulusi* venoms revealed considerable qualitative and quantitative differences. While the venoms of all three species were dominated by low-molecular-weight toxins (<10 kDa), major differences in intermediate- (20–50 kDa) and high-molecular-weight (>50 kDa) toxins were also noted (Figure 1B). Barring the minor intraspecific differences in intensities and banding patterns of the venoms of *B. sindanus* from Rajasthan and Maharashtra, distinct SDS-PAGE profiles were observed for the three species. To precisely identify the compositional differences, we subjected whole venoms to tandem mass spectrometry. Searching the generated spectra against NCBI-NR Serpentes database and the *Bungarus* transcriptomes generated in this study led to the identification of 147, 225 and 132 non-redundant toxin proteins in the venoms of *B. sindanus*, *B. caeruleus* and *B. romulusi*, respectively (Figure 3A; Appendix A). The identified toxins belonged to 19 families, namely three-finger toxin (3FTx), phospholipase A_2_ (PLA_2_), β-bungarotoxin, cobra venom factor (CVF), cysteine-rich secretory proteins (CRISP), 5′-nucleotidase, disintegrin-like toxins, C-type lectin (CTL), L-amino-acid oxidase (LAAO), nerve growth factor (NGF), Kunitz-type serine protease inhibitor (Kunitz), vespryn, acetylcholinesterase (AChE), phospholipase B (PLB), snake venom serine protease (SVSP), vascular endothelial growth factor (VEGF), phosphodiesterases (PDE), hyaluronidase (HYL), and peptidase S1 (Figure 3A; Appendix A).

Mass spectrometry of the venoms of the three krait species revealed significant differences in proportions of neurotoxic 3FTxs and PLA_2_s. While the venom proteome of *B. caeruleus* was enriched with various subtypes of neurotoxic 3FTxs (65%), surprisingly, only a smaller fraction of *B. sindanus* (14%) and *B. romulusi* (10%) venom proteomes were constituted by them (Figure 3A). In contrast, nearly 53% of the *B. sindanus* and 42% of *B. romulusi* venoms were made up of PLA_2_ toxins, whereas less than 11% of *B. caeruleus* venom contained PLA_2_s. The relative abundance of 3FTx, followed by PLA_2_ and β-bungarotoxin, has also been documented in the venom proteome of a nearby population (Tamil Nadu) of *B. caeruleus* [13]. While the relative abundance of 3FTx and PLA_2_ in *B. sindanus* (Maharashtra) was also very similar to the abundance of these components in the venom proteomes of *B. sindanus* from Pakistan [14] and Rajasthan [15], the venom of the former population was more abundant in PLA_2_ toxins (Figure 3A). Interestingly, nearly 12% of the *B. sindanus* venom proteome was composed of β-bungarotoxin: a presynaptic neurotoxin, which is a heterodimer of PLA_2_ and Kunitz peptides [16], while *B. caeruleus* and *B. romulusi* only contained around 7% and 4% of this toxin type, respectively. Quantitative proteomics experiments have previously shown that the relative abundance of this toxin type varies significantly between *Bungarus* species, ranging from nearly half of the venom composition in certain South East Asian species to minor amounts (0.6% to 5%) documented in the Sri Lankan *B. caeruleus* [14]. Astonishingly, the venom of the *B. romulusi* (37%) was found to be rich in acetylcholinesterase, while constituting a minor fraction of *B. caeruleus* (5%) and *B. sindanus* (1%) venoms. The three species also differed in their abundance of LAAO, CRISP and Kunitz serine protease inhibitors (Figure 3A).

These findings were also supported by the densitometric analyses of SDS-PAGE profiles (Figure 1B and Appendix A). The abundance of AChE (64–78 KDa), CRISP (~26 KDa), β-bungarotoxin (~16 KDa under reducing and 21 KDa under non-reducing conditions: Figure 1B and Appendix A), PLA_2_ (13–15 KDa), Kunitz (~9 KDa) and 3FTx (7–9 KDa) correlated with the intensities of the bands at the respective molecular weight ranges.

### 2.3. Comparative Transcriptomics

RNA sequencing of *B. sindanus*, *B. caeruleus* and *B. romulusi* tissues on Illumina’s HiSeq X platform resulted between 24,343,212 and 51,842,239 sequences from the venom glands and between 12,058,000 and 55,336,796 sequences from the intestinal tissues (Appendix A). De novo comparative transcriptomes assembled with these sequences retrieved 229,763, 308,570 and 225,522 transcripts from *B. sindanus*, *B. caeruleus* and *B. romulusi*, respectively. Annotation of venom gland transcripts, followed by differential expression analyses, revealed the upregulation of multiple toxin-encoding genes in the venom glands in comparison to the intestine (Appendix A). In addition to recovering large numbers of short- (Type I) and long-chain (Type II) α-neurotoxins, κ-bungarotoxin, unconventional 3FTxs, PLA_2_s and β-bungarotoxin, we identified 5′-nucleotidase, acetylcholinesterase, CRISP, CTL, disintegrin-like SVMPs, hyaluronidase, Kunitz, LAAO, natriuretic peptide, NGF, phospholipase B, VEGF and vespryn as differentially up-regulated in the venom glands in comparison to the physiological tissue (between 2 to 22 log_2_ fold change; probability > 0.9). While the role of this toxin in snakebite pathology remains unknown, dipeptidyl peptidase-4 (DPP-4) sequences were uniquely retrieved from the venom glands of *B. caeruleus* [17].

In contrast to the stark differences observed in the venom proteomes of the three species, venom gland transcriptomes were fairly similar (Figure 3B). Consistent with the venom gland transcriptomes of the other Asian kraits [18,19,20], comparative tissue expression analyses in this study revealed the domination of neurotoxic 3FTx and PLA_2_ transcripts in the venom glands of all three *Bungarus* species (Figure 3B). While between 29 to 36% of all toxin transcripts belonged to the 3FTx superfamily, PLA_2_s constituted between 23% to 29% of the venom gland toxin transcriptome. β-bungarotoxin was found equally abundant in *B. caeruleus* and *B. romulusi* (~9%), whereas 20% of the venom gland transcriptome of *B. sindanus* consisted of this toxin type. Minor differences were observed in the abundance of CRISPs, disintegrin-like SVMPs, Kunitz, LAAO and natriuretic peptides between the three species. The significant difference documented between the venom proteomes and venom gland transcriptomes, particularly in the abundance of 3FTx, PLA_2_ and β-bungarotoxin, suggests the possible role of post-transcriptional regulatory mechanisms in shaping the venoms of these cryptic kraits. Among the physiological protein-coding genes that were found to be overexpressed in the venom glands of all three species of kraits were those encoding a large number of disulphide isomerase isoforms (2.1–7.1 log_2_ fold change; probability > 0.9). Specific isoforms of these enzymes, which are essential for the proper folding of cysteine-rich proteins, including venom toxins, have also been shown to be overexpressed in the venom producing cells of other venomous animals [21].

### 2.4. Coagulopathic Effects of Bungarus Venoms

While this is not a major pathology documented in Elapidae snake envenoming, in the preclinical setting, the venoms of certain Indian elapids have been shown to affect hemostasis by interfering with the intrinsic and/or extrinsic coagulation cascades [15,22,23,24]. Therefore, we examined the abilities of *Bungarus* venoms to affect these pathways via activated partial thromboplastin time (aPTT) and prothrombin time (PT) assays, respectively. While none of the tested venoms affected the extrinsic coagulation cascade, all venoms exhibited significant effects on the intrinsic pathway (Appendix A). In line with the previous findings [15], the venoms of *B. sindanus* and *B. caeruleus* from Maharashtra were found to significantly alter aPTT values (coagulation delayed by 111 and 600 s at 40 µg, respectively), whereas *B. romulusi* venom exhibited considerable effects only at very high concentrations (coagulation delayed by 73 s at 40 µg). Similarly, in the haemolytic assays, *B. caeruleus* exhibited maximal activity in comparison to the 0.5% Triton X-100 positive control (29%), closely followed by *B. sindanus* (16%), whereas *B. romulusi* showed minimal haemolytic effects (8%) (Appendix A).

### 2.5. In Vitro Venom Recognition Potential of Commercial Antivenoms

We performed enzyme-linked immunosorbent assay (ELISA) and immunoblotting in vitro experiments to measure the potential of commercial Indian polyvalent antivenoms (Premium Serums and Vaccines Pvt. Ltd. (PSVPL), Pune, India and Haffkine BioPharmaceutical Corporation Ltd. (Haffkine), Mumbai, India) in recognising the venoms of the three krait species (Figure 4; Appendix A). While both antivenoms recognised the venoms of *B. caeruleus* (titre: 1:500; PSVPL: 1.02 Optical Density (OD); Haffkine: 0.89 OD) and *B. sindanus* (titre: 1:500; PSVPL: 0.89 OD; Haffkine: 0.76 OD) relatively well, they exhibited limited to poor recognition potential against the venom of *B. romulusi* (titre: 1:500; PSVPL: 0.24 OD; Haffkine: 0.25 OD) (Figure 4). These results are indicative of the poor preclinical effectiveness of Indian antivenoms against *B. romulusi*.

Immunoblotting experiments revealed that the commercial antivenoms very poorly recognised the highly abundant low-molecular-weight toxins (<20 KDa) in *Bungarus* venoms (Figure 5 and Appendix A). While Premium serums antivenom exhibited relatively increased binding towards the *B. caeruleus* venom in comparison to the venoms of *B. romulusi* and the two populations of *B. sindanus*, the Haffkine antivenom failed to recognise several toxins from all *Bungarus* species tested in the study (Figure 5 and Appendix A). As the Haffkine antivenoms are manufactured by sourcing venoms from the state of Maharashtra in Western India, the poor venom recognition of this antivenom against the source venom immunogen population is surprising. Moreover, non-specific binding of naive horse Immunoglobulins G (IgGs) towards certain toxin bands was also observed.

### 2.6. Toxicity Profiles

Evaluation of toxicity profiles (or the median lethal dose (LD_50_)) of *Bungarus* species in the mouse model revealed dramatic differences in venom potencies. While the toxicity of *B. caeruleus*, the most potent of the ‘big four’ Indian snakes, was in line with previous findings (0.25 mg/kg; [15]), the newly discovered *B. romulusi* exhibited six times more toxicity (0.045 mg/kg). Moreover, *B. sindanus* (the 17 DSR krait from Maharashtra) was found to be amongst the most toxic snakes identified in India. Consistent with the toxicity score reported for the Rajasthani population of this species (0.018 mg/kg; [15]), *B. sindanus* from Maharashtra was characterised with an LD_50_ of 0.02 mg/kg (Figure 6A, Appendix A).

### 2.7. In Vivo Venom Neutralisation

Consistent with the outcomes of in vitro venom binding assays, in vivo venom neutralisation experiments in the murine model revealed the poor preclinical potencies of commercial antivenoms against *B. sindanus* and *B. romulusi*. While the tested antivenom (PSVPL) exhibited relatively better performance against *B. caeruleus* from Maharashtra (0.744 mg/mL) than the marketed value (0.45 mg/mL), it was characterised with alarmingly inefficient neutralisation potencies against both *B. sindanus* (0.15 mg/mL) and *B. romulusi* (0.13 mg/mL) (Figure 6B, Appendix A). This trend remains consistent with previous findings where the commercial polyvalent antivenom manufactured by VINS performed relatively better against the venom of *B. caeruleus* from India (0.48 mg/mL) but performed poorly against *B. sindanus* from Pakistan (0.25 mg/mL) [14,25]. Interestingly, the PSVPL antivenom was previously found to exhibit very poor neutralisation efficacy against the venoms of both *B. sindanus* from Rajasthan (0.272 mg/mL) and *B. caeruleus* from Punjab (0.312 mg/mL) [15].

## 3. Discussions

### 3.1. Cryptic Kraits of Southern and Western India

India is an abode to at least seven described species of kraits [7], some of which possess the most potently toxic venoms in the world [15]. To date, kraits with 15, 17 and, very rarely, 19 DSR have been reported in Western India. While kraits with 15 DSR in Peninsular India are identified as *B. caeruleus*, those with 17 or 19 DSR are attributed to either *B. sindanus* or *B. sindanus walli* (some authors have also called the latter *B. walli*)**, depending on their geographical distribution. Kraits with 17 DSR in India that are geographically closer to the Sind region of Pakistan are believed to be *B. sindanus*, whereas isolated records of kraits with 17 or 19 DSR in the other parts of the country have been called *B. sindanus walli* or *B. walli*, without any scientific evidence. In fact, Wall’s krait (*B. walli*) was originally described by a British herpetologist, Frank Wall, from the princely State of Oudh, Faizabad [10], which was later synonymised with the Sind krait (*B. sindanus*) and then assigned a subspecies status (*B. sindanus walli*) by Khan [6]. Given their phenotypic similarity (Figure 7), it can be challenging for clinicians and laypeople to distinguish them from one another.

Phylogenetic reconstructions using cytochrome *b* and ND4 mitochondrial markers in this study revealed fascinating insights into the evolution of cryptic kraits in Southern and Western India. Consistent with previously unpublished suggestions [4], our analyses did not recover any support for describing the 17 mid-body DSR kraits in Western India as a subspecies of *B. sindanus* (i.e., *B. sindanus walli*), or a completely distinct species (*B. walli*), but rather unravelled the limited molecular divergence of this population in comparison to *B. sindanus* in Pakistan (0.32% and 3.03% for ND4 and cyt *b* markers, respectively; Appendix A). Considering this compelling molecular evidence, we suggest renaming the 17 DSR kraits in Western India to *B. sindanus*.

Fascinatingly, phylogenetic analyses revealed that the 15 DSR krait lineage is highly polyphyletic, with the Karnataka population forming a completely distinct clade (BPP of 1 for cyt *b* and ND4). Considering the very high molecular divergence of this clade from the other 15 DSR kraits (6.6% to 13.21% and 6.96% to 8.61% for cyt *b* and ND4 markers, respectively), we name this lineage as *B. romulusi* in the honour of the famous Indian herpetologist, Romulus Whitaker. While we did not document significant morphological differences between *B. caeruleus* and *B. romulusi*, with the exception of the colour, which by itself is an unreliable character, we had access to a limited number of samples due to restrictions in our permissions from the authorities. The description of this species is, therefore, based on the overwhelming support drawn from phylogenetic reconstructions, sequence divergence estimates, venom proteomes, and toxicity profiles. Interestingly, while a minimal sequence divergence was observed in the ND4 and many cyt *b* sequences of the 15 DSR kraits in Maharashtra, Orissa (cyt *b*: 3.42) and *B. caeruleus* in Pakistan (ND4: 2.35% to 3.73% and cyt *b*: 3.69% to 3.93%), larger differences were also observed in the cyt *b* region of the other individuals (5.61% to 6.55%). The significant divergence observed in the cyt *b* marker, perhaps, highlights the existence of additional unrecognised species and/or genetically divergent populations in this clade. However, in-depth investigations involving broader sampling are necessary to precisely delineate the range distribution and biogeographic histories of various krait species in the Indian subcontinent.

### 3.2. The Clinical Impact of Bungarus Venoms

Kraits are amongst the deadliest snakes in India and are responsible for nearly 18% of the annual snakebite cases [2]. In addition to the severe neurotoxic symptoms that are often documented in krait bite victims, atypical symptoms have also been previously reported [4]. The venom of *B. sindanus* was theorised to contain large amounts of myotoxins and/or cardiotoxins as myocardial damages were documented in a young bite victim in Maharashtra, who did not have a previous history of heart disease [4]. While we did not detect any cardiotoxins, the venom of *B. sindanus* was dominated by PLA_2_ toxins (Figure 3), some of which have been shown to exhibit highly myotoxic activities [26,27,28]. Interestingly, trace amounts of cardiotoxins were detected in the venom of *B. caeruleus* from this region. Moreover, the administration of neostigmine in the aforementioned bite victim of the Sind krait was reported to be ineffective [4]. These findings are consistent with the activities of the β-bungarotoxin, which constituted a large portion of the venom of *B. sindanus* in these regions. As β-bungarotoxins destroy the motor nerve terminals, deplete synaptic vesicles, and cause axonal degeneration and flaccid paralysis [29], the administration of neostigmine has been shown to be largely ineffective [30]. In parallel to the prolonged neuromuscular paralysis caused by the action of the presynaptic β-bungarotoxin, α-neurotoxin in krait venoms can target postsynaptic neuronal acetylcholine receptors (nAChR) at the neuromuscular junction [31]. The venom gland transcriptome of *B. caeruleus* was particularly rich in both Type-I (short) and Type-II (long) α-neurotoxin transcripts (62% of all 3FTxs), and a large proportion of these (66%) constituted the venom proteome as well. While the venom glands of *B. sindanus* and *B. romulusi* also transcribed large numbers of α-neurotoxins (65% and 87% of all 3FTx transcripts), their venoms were surprisingly constituted by a minor fraction of these proteins (14% and 10%, respectively). Similarly, structurally complex κ-bungarotoxins, which are exclusively found in *Bungarus* spp., can antagonise neuronal nAChRs, resulting in flaccid paralysis [32]. While a large proportion of κ-bungarotoxin transcripts were recovered from the venom gland of *B. caeruleus* (20%), only minor amounts were detected in *B. romulusi* (1.63%) and were completely absent in the *B. sindanus* transcriptome (Appendix A).

AChE is yet another important *Bungarus* venom component that is involved in the rapid hydrolysis of acetylcholine at the neuromuscular junction [33]. Previous studies have reported varied proportions of AChE, ranging from as low as 0.02% to 12.6% from closely related *Bungarus* spp. [34,35]. While *Bungarus* venom AChEs possess substrate specificity and high catalytic activity [36], they did not exhibit pronounced toxicity or synergy when tested using in vivo models [37]. However, the detection of a significantly higher abundance of AChE in *B. romulusi* venoms might insinuate its potential role in snakebite envenoming and warrants further investigation.

### 3.3. The Role of Ecology, Environment and Phylogenetic Histories in Driving Snake Venoms

Ecology and environment are well-known to influence the composition and potencies of snake venoms [38,39,40,41,42]. Differing biogeographies have been shown to have a significant impact on the venoms of *Naja naja* and *Daboia russelii*, two of the medically most important ‘big four’ snakes [43,44,45,46,47,48,49]. However, despite sharing overlapping distribution ranges, at least in some parts of Western India (the precise range overlap is yet to be fully understood), the venom profiles of the investigated cryptic kraits were surprisingly distinct. Even though the differences were mainly in terms of the proportion of 3FTxs and PLA_2_s, very distinct toxicity profiles were documented. Consistent with the previous findings [15], *B. sindanus* venom was extremely toxic to mice, being over 11 times more potent than *B. caeruleus*, which makes it the most toxic snake in the Indian subcontinent and amongst the most toxic snakes in the world. It should be noted that this species has been reported to show a preference for mammalian prey animals over reptiles [4], which could explain the extremely high toxicity of this venom in the mouse model. The venom of *B. romulusi* was also found to be a lot more potent than that of *B. caeruleus*. When tested in the murine model, this species was found to be six times more potent than *B. caeruleus*. Considering the overlapping distribution ranges of these closely related species, the outcomes of toxicity profiling experiments are surprising. While subtle differences in prey-preference, diet breadth and predator pressures cannot be ignored, these results are, perhaps, also suggestive of the roles of phylogenetic histories in driving snake venom compositions and toxicity profiles.

### 3.4. Post-Genomic Regulation Shapes Krait Venom Profiles

Comparative tissue transcriptomics revealed that the venom glands of all three krait species in Southern and Western India were highly similar in their toxin compositions. The venom glands of *B. sindanus*, *B. caeruleus* and *B. romulusi* were highly enriched with PLA_2_s and various functional subtypes of 3FTxs. The only considerable documented difference was with respect to the abundance of β-bungarotoxin in the venom gland transcriptome of *B. sindanus* (20%), whereas those of the other two species had similar amounts of CRISP transcripts (10–13%). In contrast to their highly similar venom gland transcriptomes, and despite sharing overlapping range distributions (at least in the case of *B. caeruleus* and *B. sindanus*) and, consequently, very similar environments, the venom proteomes of these kraits were considerably distinct. While the venom of *B. caeruleus* was still enriched with neurotoxic 3FTxs (66%), *B. sindanus* (52%) and *B. romulusi* (42%) venoms were largely composed of PLA_2_s. In contrast to this switch from a combination of 3FTx and PLA_2_-enriched transcriptome to a largely PLA_2_-dominated venom proteome in the Sind krait, the abundance of β-bungarotoxin remained nearly the same (~12%). The abundance of this toxin type significantly reduced from nearly 10% of the venom gland transcriptome to 4% of the venom proteome in *B. romulusi*. β-bungarotoxin has been shown to be responsible for the fatally toxic effects accompanying *Bungarus* envenomings [50]. However, the precise role of this toxin type in driving the extreme potency of *B. sindanus* remains to be understood. Interestingly, differences between the expressed transcriptomes and venom proteomes have also been reported in other *Bungarus* spp. in Southeast Asia [20]. However, these changes were attributed to post-translational modifications by the authors, who argued that this might be a result of the differences in the amounts of protein processing enzymes. Since microRNAs (miRNA) have been previously documented to facilitate ontogenetic shifts in snake venom compositions via post-genomic mechanisms [51], future investigations into the miRNA profiles of Indian kraits are warranted to unravel their roles in shaping these contrasting venom profiles.

### 3.5. The Need for Regionally Effective Antivenoms

While the underlying mechanisms driving the evolution of venom remain to be elucidated, the impact of this variation on the preclinical effectiveness of snakebite therapy could not be any clearer. The effectiveness of the PSVPL antivenom in neutralising the fatal effects of *B. caeruleus* venom (0.74 mg/mL), which is used in the manufacturing process, was well over the marketed neutralising potency (0.45 mg/mL). Surprisingly, despite exhibiting a venom recognition potential that is very similar to the binding against *B. caeruleus*, the antivenom performed extremely poorly against the venom of the closely-related *B. romulusi* (0.13 mg/mL). However, consistent with the results for the Rajasthan population (0.27 mg/mL) [15], the commercial antivenom performed extremely poorly against the venom of *B. sindanus* from Maharashtra (0.15 mg/mL). In addition to their inability to counter interspecific venom variation, Indian antivenoms have also been shown to be inefficacious in neutralising the intraspecific variation in the venoms of distant *B. caeruleus* populations [15,25].

These findings very clearly demonstrate the inadequacy of the polyvalent antivenom in conferring cross-neutralisation even against species that are closely related to the ‘big four’ snakes. Considering the remarkable numbers of medically important snake species in the country, and the diversity of toxin types they secrete, it would be practically difficult to produce a single antivenom product. Moreover, the preclinical outcome of this product is unlikely to improve, as it would be unreasonable to expect the required amounts of neutralising antibodies against the diversity of venoms of the pan-Indian snake species/populations. This is especially true for an antivenom production strategy that has remained virtually unchanged over the past century. While significant efforts are being put into producing recombinant antivenoms with broadly neutralising antibodies that would offer a very effective clinical alternative, they are far from fruition. Therefore, establishing regional venom collection centres in strategically chosen locales across the country, determined by the outcomes of phylogenetic and venomics research, will enable the development of regionally effective antivenoms that are likely to confer protection against the medically important snakes by region. Thus, there is a pressing need to produce regionally effective antivenoms in India to safeguard the lives of hundreds of thousands of snakebite victims.

## 4. Conclusions

In conclusion, phylogenetic reconstructions of evolutionary histories of phenotypically similar cryptic kraits in Southern and Western India provided fascinating insights. Bayesian and maximum-likelihood analyses using mitochondrial markers revealed a new species of krait in Southern India, which we named Romulus’ krait (*B. romulusi*). These analyses also revealed that the krait with 17 mid-body dorsal scale rows in Western India is not a subspecies of the Sind Krait (*B. sindanus walli*) or a distinct species (*B. walli*) as previously believed but was rather genetically very similar to *B. sindanus* in Pakistan. Moreover, venom proteomics and comparative tissue transcriptomics experiments unravelled completely contrasting venom profiles in each krait species. While the venom proteomic composition differed significantly between the three species, their transcriptomes were largely similar, highlighting the possible role of post-genomic regulatory mechanisms in shaping their venoms. In vitro and in vivo experiments, assessing the effectiveness of commercial Indian antivenoms in countering these distinct krait venoms, revealed a severe shortcoming. While the antivenom preclinically neutralised *B. caeruleus* venom effectively, its efficiency against the venoms of *B. sindanus* and *B. romulusi* was severely limited. Taken together, our findings highlight the importance of phylogenetic studies in identifying medically important species/populations and the urgent need for regionally-effective antivenoms in India.

## 5. Material and Methods

### 5.1. Sampling Permits, Snake Venoms and Antivenoms

One adult female snake of each species was wild-caught from the Pune district in Maharashtra with due approval from the Maharashtra State Forest Department (Desk-22 (8)/WL/Research/CR-60 (17-18)/2708/2018-2019) (Figure 1A). In addition, a female krait was also caught from Southern Karnataka with appropriate permission from the Karnataka State Forest Department (PCCF(WL)/E2/CR06/2018-19). Individually collected venoms were flash-frozen immediately, lyophilised and stored at −80 °C until further use. Post venom extraction, snakes were housed for three days in captivity and venom glands were collected on the fourth day. Snakes were humanely euthanised with a single intracardiac dose of sodium pentobarbital (100 mg/kg), followed by the surgical extraction of venom glands and other physiological tissues. Tissue samples were immediately flash-frozen and stored at −80 °C until use. The Bradford assay [52] was used to determine the protein concentrations of venom and commercial antivenoms with Bovine Serum Albumin (BSA) and Bovine Gamma Globulin (BGG) as standards, respectively (Appendix A). Additional details of venom and antivenom samples investigated in this study are provided in Appendix A, respectively. Given that the *Bungarus* spp. are protected in India under the Schedule IV of the Wildlife (Protection) Act (1972), we were only granted permission by the Karnataka Forest Department to euthanise a single individual for venom gland transcriptomics (under the purview of licensed ‘hunting and killing’). Tissue samples, including the heart tissue that was used for DNA isolation, have been deposited with the Bombay Natural History Society (BNHS) as paratypes: heart (BNHS 3611) and intestine (BNHS 3612). However, in case any of the live specimens under captivity suffer death by natural causes, an intact specimen will be housed as a topotype in the museum at the Centre for Ecological Sciences, IISc, Bangalore (CES-BNHS 3611-02), and additional specimens (if and when available) will be deposited to BNHS with due permissions from the Forest Department. Full-body images of *B. romulusi* have been submitted to MorphoBank (#3897).

### 5.2. Ethical Clearances

The toxicity profiles of the venom samples and the neutralisation potencies of commercial Indian antivenoms were estimated in the murine model, as per the World Health Organisation (WHO)-recommended protocols. Experiments were conducted on male CD-1 mice (18–22 g) after acquiring due approval from CPCSEA and IAEC: CAF/Ethics/770/2020 (approval date: 16 October 2020). Throughout the course of the experiment, animals were housed in the Central Animal Facility at IISc (48/GO/ReBi/SL/1999/CPCSEA; 11-03-1999). The ability of *Bungarus* venoms in inducing coagulopathies in human blood was evaluated after obtaining necessary approvals from the Institutional Human Ethics Committee (IHEC No: 5-24072019; approval date: 24 July 2019), IISc.

### 5.3. DNA Isolation and Sequencing

Genomic DNA was isolated from the heart muscle tissue using the Xpress DNA Tissue kit following the manufacturer’s recommendations (MagGenome, Leeds, UK). The quality of the isolated DNA was evaluated by estimating the absorbance ratio at 260 nm and 280 nm wavelengths in an Epoch 2 microplate spectrophotometer (BioTek Instruments, Inc., Winooski, VT, USA) and visualisation on 1% agarose gel electrophoresis. Polymerase Chain Reaction (PCR) was performed on a ProFlex PCR System (Thermo Fisher Scientific, Waltham, MA, USA) to amplify specific regions of two mitochondrial markers (ND4 and cyt *b*) using universal primers [53,54] (Appendix A). PCR reaction mixtures (50 μL) contained 1 μL of template DNA (~50 ng), 25 μL of Taq DNA Polymerase Mastermix (Tris-HCl pH 8.5, (NH_4_)_2_SO_4_, 3 mM MgCl_2_, 0.2% Tween 20, 3 μL of 25 mM MgCl_2_, 0.4 mM dNTPs, Amplicon Taq polymerase), 2.5 μL each of forward and reverse primers and 19 μL of nuclease-free water. For PCR amplification, the following thermal profiles were used: initial denaturation at 94 °C for 5 min, then 35 cycles of denaturation (94 °C for 30 s), annealing (T_A_ for 35 s) and extension (72 °C for 2 min), followed by a final step of extension for 10 min at 72 °C. Following gel purification using the QIAquick PCR and Gel Cleanup Kit (Qiagen, Hilden, Germany), amplicons were sequenced on the Applied Biosystems 3730xl platform (Thermo Fisher Scientific, Waltham, MA, USA) and the sequence data were acquired with Sequence Scanner Software v2.0. Sequences generated in this study have been deposited to the GenBank database, and the data along with accession numbers have been provided in Appendix A.

### 5.4. Phylogenetic Reconstructions

Nucleotide datasets were assembled for both genes by retrieving additional sequences from NCBI’s GenBank repository. Sequences were aligned using Clustal-Omega [55]. For reconstructing the phylogenetic histories of *Bungarus* spp., we first used the ML-based approach implemented in the PhyML package [56]. The best nucleotide substitution model for reconstruction was identified using the Smart Model Selection (SMS) tool on the ATGC server [57] with the Subtree-Pruning-Regrafting (SPR) method for searching tree topologies and 100 bootstrapping replicates for the evaluation of node support. The Bayesian inference-based phylogeny was also estimated using MrBayes 3.2.7 [58,59]. The analysis was executed on four independent runs, each with nine parallel Markov chain simulations, running for 10 million generations or until the convergence of chains (i.e., 0.01 standard deviation of split frequencies). The trees and the corresponding parameter estimates were sampled every 100th generation, of which, 25% were discarded as burn-in. A majority-rule consensus tree and the posterior probability of each node were estimated using trees selected post-burn-in. We further estimated the evolutionary divergence between ND4 and cyt *b* sequences using the Maximum Composite Likelihood model [60] in MEGA X [61] with 500 bootstrap replicates. Additionally, *p*-distances were also calculated in MEGA X [61].

### 5.5. Gel Electrophoresis

Electrophoretic separation of venom samples was accomplished on SDS-PAGE. Briefly, venoms normalised for protein content (12 µg) were resolved in a 12.5% gel under reducing conditions, followed by staining with Coomassie Brilliant Blue R-250 (Sisco Research Laboratories Pvt. Ltd., Mumbai, MH, India) and imaging under an iBright CL1000 (Thermo Fisher Scientific, Waltham, MA, USA) gel documentation system. The densitometric analyses of the gel bands were performed using the ImageJ software [62].

### 5.6. Liquid Chromatography-Tandem Mass Spectrometry (LC-MS/MS)

Crude snake venom samples (50 µg) in 25 mM ammonium bicarbonate were reduced with 10 mM dithiothreitol for 30 min at 37 °C, alkylated with 100 mM iodoacetamide for 30 min in the dark and digested with trypsin (0.2 µg/µL) at 37 °C overnight, in a total volume of 50 µL. The reaction was stopped using 0.1% formic acid, and desalting was carried out to remove the buffer salt and detergent using ZipTip with acetonitrile. The resulting tryptic peptide mixtures were separated using the Thermo EASY nLC 1200 series system (Thermo Fisher Scientific, Waltham, MA, USA) coupled online with a Thermo Orbitrap Fusion^TM^ Mass Spectrometer (Thermo Fisher Scientific, Waltham, MA, USA). We loaded 0.5 µg of peptide mixture on a 50 cm × 75 µm C18 (3 µm, 100 Å) nano-LC column. The mobile phase solution consisted of 0.1% formic acid in HPLC grade water (solution A) and an elution buffer of 0.1% formic acid in 80% acetonitrile (solution B). The flow rate of solution B was set to 300 nL/min and was used in the following concentrations: 10–45% over 98 min, 45–95% over 4 min and 95% over 18 min. MS spectra were acquired in positive mode at 2.0 kV, with a capillary temperature of 200 °C, using 1μ scan in the range 375–1700 m/z, maximum injection time of 50 ms and resolution of 120,000. Fragment scans (MS/MS) were performed using an ion trap detector with high collision energy fragmentation (30%), scan range between 100–2000 m/z and maximum injection time of 35 ms. For the identification of various toxin families in the proteomic profiles of venoms, raw MS/MS spectra were searched against the National Center for Biotechnology Information’s (NCBI) non-redundant (nr) database (Serpentes: 8570; November 2020), as well as the venom gland transcriptomes generated in this study, using PEAKS Studio X Plus (Bioinformatics Solutions Inc., Waterloo, ON, Canada). Parent and fragment mass error tolerances were set at 10 parts per million (ppm) and 0.6 Da, respectively. Cysteine carbamidomethylation was set as fixed modification while methionine oxidation and deamidation of asparagine or glutamine were set as variable modifications. A maximum of 2 missed cleavages by trypsin in the semispecific mode were allowed. Filtration parameters for match acceptance were set to FDR 0.1%, detection of ≥1 unique peptide and −10lgP protein score ≥40. Hits with at least one unique matching peptide were considered for downstream analyses. Mass spectrometry data have been deposited to the ProteomeXchange Consortium via the PRIDE [63] partner repository, with data identifier: PXD023127. The relative abundance of each toxin hit in a sample was determined by its area under the spectral intensity curve (AUC) obtained from PEAKS Studio analyses, relative to the total AUC for all toxin hits. The relative abundance of a toxin hit (X) was calculated as follows:$$Relative\;abundance\;of\;toxin\;hit\;X\;\left(\%\right)\;=\frac{AUC\;of\;toxin\;hit\;X}{Total\;AUC\;of\;all\;toxin\;hits}\times 100$$

The results of mass spectrometry analyses are provided in Appendix A.

### 5.7. RNA Isolation, Library Construction and RNA Sequencing

Total RNA was isolated from venom glands and physiological tissues of freshly euthanised snakes using the TRIzol™ Reagent (Invitrogen, Thermo Fisher Scientific, Waltham, MA, USA) according to the manufacturer’s protocol (Appendix A). Contaminant DNA was removed from the isolated RNA samples by treatment with Turbo DNase (Thermo Fisher Scientific, Waltham, MA, USA), followed by another round of extraction with the TRIzol™ Reagent to remove the DNase enzyme. Quantification of RNA was performed using an Epoch 2 microplate spectrophotometer (BioTek Instruments, Inc., Winooski, VT, USA) and Qubit Fluorometer (RNA High Sensitivity kit: Cat# Q32852; Agilent Technologies, Santa Clara, CA, USA), while the integrity was assessed on the Agilent 4200 TapeStation system using RNA HS ScreenTape (Cat# 5067-5579; Agilent Technologies, Santa Clara, CA, USA). With the exception of a physiological tissue RNA (RIN: 7), samples with an RNA Integrity Number (RIN) of ≥8 were down-selected for RNA sequencing (Appendix A).

Prior to the library preparation, the ribosomal RNA (rRNA) was removed with the help of NEBNext Poly(A) mRNA Magnetic Isolation Module, followed by fragmentation of the selected mRNA in the presence of divalent cations at elevated temperatures. Library preparation was then performed using the NEBNext^®^ Ultra™ RNA Library Prep Kit for Illumina^®^, following the manufacturer’s protocol. The resultant cDNA libraries were amplified by subjecting them to 13 cycles of PCR. Following the purification of PCR products, qualitative and quantitative assessments were performed using Qubit High Sensitivity Assay (Cat# Q32852; Agilent Technologies) and D1000 DNA ScreenTapes (Cat# 5067-5582; Agilent Technologies) on a TapeStation, respectively. The resultant libraries were sequenced on the HiSeq X System (Illumina, San Diego, CA, USA) to generate at least 50 million paired-end (2 × 150 bp) reads per sample. The raw sequencing data generated in this project has been deposited to the Sequence Read Archive (SRA) at NCBI (Bioproject: PRJNA681550; SRA: SUB8677374, SUB8648213 and SUB8655883).

### 5.8. Quality Filtering, Transcriptome Assembly and Transcript Annotation

Prior to the transcriptome assembly, Illumina’s basespace pipeline was used for de-multiplexing and filtering high-quality sequencing reads. This was followed by quality filtering steps in Trimmomatic v0.39 [64], where the adapter sequences, leading and trailing low-quality bases (<3), short reads (<20 bases) and low-quality reads (<25; sliding window 4) were removed. The quality of FASTQ sample files before and after trimming was validated using FASTQC v0.11.9 [65]. Quality-filtered reads were then de novo assembled into contigs using Trinity v2.11.0 [66] with the following parameters: k-mer = 25, minimum k-mer coverage = 1, minimum contig length = 200, pair distance = 500 and the maximum number of reads per graph = 200,000. The quality of the assembled transcriptome was evaluated by aligning reads back onto the transcriptome using BowTie v2.4.2 [67]. This was followed by the prediction of coding regions in transcripts that encode a minimum of 30 amino acids using TransDecoder v5.5.0 [68]. These coding regions were then annotated by performing BLAST searches [69] against the NCBI-nr database (November 2020; Serpentes (taxid: 8570)).

### 5.9. Transcript Quantification and Differential Expression Analyses

Following the alignment of reads back onto the transcriptome with BowTie v2.4.2 [67], alignment-based transcript abundance quantification, expressed in fragments per kilobase of exon per million fragments mapped (FPKM), was performed using RNA-Seq by the Expectation-Maximization (RSEM v1.3.3) tool [70]. Pairwise differential expression analysis was performed on the RSEM normalised count data using a novel nonparametric approach implemented in the NOISeq R package [71,72]. Genes were identified as differentially expressed when they met a predefined fold-change (≥2) and statistical probability (*p* ≥ 0.9) cutoffs.

### 5.10. Venom Induced Coagulopathy

The abilities of *Bungarus* venoms in altering blood coagulation cascades by affecting the two major coagulation pathways (i.e., intrinsic and extrinsic pathways) were evaluated by performing activated partial thromboplastin time (aPTT) and prothrombin time (PT) tests, respectively. Briefly, whole blood collected from healthy male volunteers was centrifuged at 3000× *g* for 10 min at 4 °C to separate red blood cells (RBC) and collect the platelet-poor plasma (PPP). Four concentrations of *Bungarus* venoms (5 µg, 10 µg, 20 µg and 40 µg) were mixed with calcium thromboplastin and activated cephaloplastin reagent with calcium dichloride (CaCl_2_) (Liquicelin-E; Tulip diagnostics, Mumbai, MH, India) to measure PT and aPTT values, respectively. Hemostar XF 2.0 coagulometer (Tulip Diagnostics, Mumbai, MH, India) was used to measure the time taken for the formation of the first fibrin clot.

### 5.11. Haemolytic Assay

The ability of *Bungarus* venoms in destroying RBCs was assessed following a previously described protocol [15,73]. Briefly, RBCs from freshly collected human blood were separated and washed five times using 1X PBS buffer (pH 7.4). The RBC pellet obtained was resuspended gently in PBS at a ratio of 1:10 and incubated with various concentrations of venoms at 37 °C. Following a 24 h incubation, the supernatant was collected after centrifugation at 3000× *g* for 10 min at 4 °C. The absorbance of the supernatant measured at 540 nm was used to calculate the haemolytic activity of the venoms relative to 0.5% Triton X (positive control). The activities of all the samples were assessed in triplicates.

### 5.12. Enzyme-Linked Immunosorbent Assay (ELISA)

The in vitro binding abilities of commercial Indian antivenoms against the *Bungarus* venoms were determined using indirect ELISA as described before [15,74]. We coated 96-well immunoplates with crude venoms (100 ng) diluted in the carbonate buffer (pH 9.6), followed by overnight incubation at 4 °C. The plates were washed six times with Tris-buffered saline (0.01 M Tris pH 8.5, 0.15 M NaCl) and 1% Tween 20 (TBST), followed by the addition of blocking buffer (5% skimmed milk in TBST). After the completion of three hours of incubation at room temperature, the plates were again washed with TBST (*n* = 6), followed by the addition of various dilutions of commercial Indian antivenoms (1:4 to 1:312500). The plates were then incubated overnight at 4 °C. Unbound primary antibodies were washed off with a series of TBST washes (*n* = 6) the next day, followed by the addition of 100 µL horseradish peroxidase (HRP)-conjugated, rabbit anti-horse secondary antibody (Sigma-Aldrich, St. Louis, MO, USA; 1:1000 dilution in PBS buffer). The plates were incubated for two hours at RT and 2,2′-Azino-bis(3-ethylbenzothiazoline-6-sulfonic acid) substrate solution (Sigma-Aldrich, St. Louis, MO, USA) was added for colour development. The absorbance, which is directly proportional to the venom recognition capability of the antivenom, was measured at 405 nm in an Epoch 2 microplate spectrophotometer (BioTek Instruments, Inc., Winooski, VT, USA). The statistical significance of the result was evaluated by two-way ANOVA with Tukey’s and Dunnett’s multiple comparisons in GraphPad Prism (GraphPad Software 8.0, San Diego, CA, USA, www.graphpad.com). IgG from an unimmunised horse was used to calculate the non-specific binding cut off as described previously [15,74].

### 5.13. Immunoblotting

To measure the *Bungarus* venom recognition potential of commercial Indian antivenoms, we performed immunoblotting experiments [15,74]. Following the separation of venom proteins on a 12.5% SDS-PAGE gels, proteins were transferred onto a nitrocellulose membrane using the Trans-Blot Turbo Transfer System (BioRad, Hercules, CA, USA) as per the instructions provided by the manufacturer. The reversible Ponceau S stain was used to confirm transfer efficacy, and then the membrane was blocked with a blocking solution (5% skimmed milk in TBST) at 4 °C overnight. On the next day, the membrane was washed with TBST six times and incubated overnight with a fixed concentration of commercial antivenom (1:200) at 4 °C. On the third day, after a series of TBST washes (*n* = 6), the membrane was incubated with HRP-conjugated rabbit anti-horse secondary antibody at 1:2000 dilution for two hours. Finally, an enhanced chemiluminescent substrate was used to visualise the blotting results in an iBright CL1000 gel imaging system (Thermo Fisher Scientific, Waltham, MA, USA). The densitometry analyses of the bands were performed using ImageJ [62].

### 5.14. The Intravenous Median Lethal Dose (LD_50_)

To evaluate the venom potencies of *Bungarus* species, WHO-recommended murine assays [75] were conducted and LD_50_ value (or the amount of venom required to kill 50% of the test population) was determined. Briefly, various concentrations (*n* = 5) of crude venom samples in physiological saline (0.9% NaCl) were injected intravenously into the caudal vein of male CD-1 mice (200 µL/mouse). The number of dead and surviving mice were recorded 24 h post venom administration and the LD_50,_ and 95% confidence intervals were calculated using Probit statistics [76].

### 5.15. The Median Effective Dose (ED_50_) of Antivenom

The preclinical efficacies of commercial Indian antivenoms in neutralising the fatal effects of *Bungarus* venoms were determined by estimating the ED_50_ value, which represents the minimum amount of antivenom required to protect 50% of the test population injected with the lethal dose of venom [75]. To greatly reduce the number of animals sacrificed in these experiments, only the antivenom manufactured by Premium Serums, which exhibited better venom recognition potential in comparison to the Haffkine antivenom, was tested in these experiments. Various volumes of the antivenom, incubated for 30 min at 37 °C with a predefined challenge dose of venom (5X LD_50_), were intravenously administered into male CD-1 mice (18–22 g; 5 mice per group). ED_50_ values and 95% confidence intervals were calculated using Probit statistics based on the number of surviving and dead mice 24 h post-venom-antivenom administration [76]. The neutralisation antivenom potencies were calculated as defined previously [15,77,78].

## Figures and Tables

**Figure 1 toxins-13-00069-f001:**
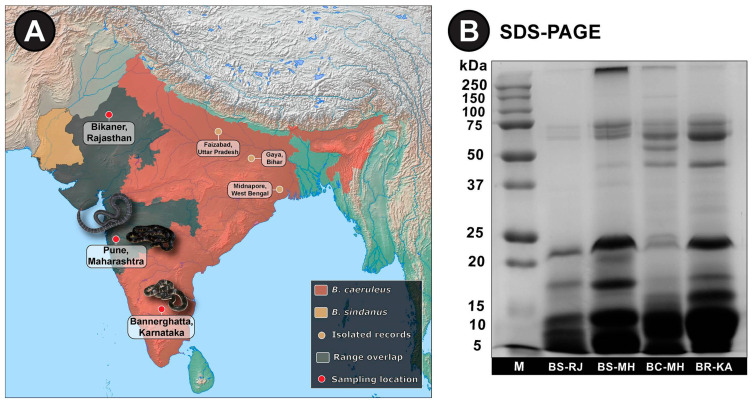
The range distribution of *Bungarus* spp. in Southern and Western India and SDS-PAGE profiles of their venoms. (**A**). Panel A of this figure shows the range distribution of *B. sindanus* (light brown) and *B. caeruleus* (red), and their range overlaps (grey) in the Indian subcontinent. Isolated records of *B. sindanus* (light brown circles), along with sampling locations (red circles) of venoms and venom glands have also been shown. (**B**). SDS-PAGE profiles of venom samples [*B. sindanus* (Rajasthan and Maharashtra), *B. caeruleus* (Maharashtra) and *B. romulusi* (Karnataka)], along with the marker (M), are shown in panel B.

**Figure 2 toxins-13-00069-f002:**
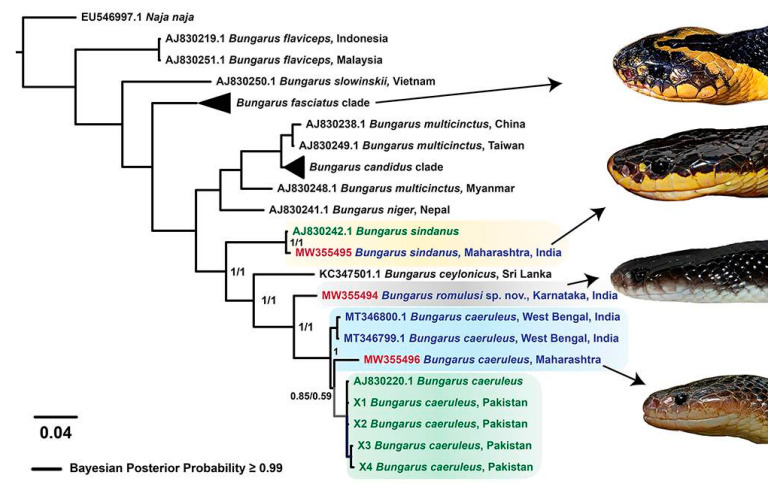
Bayesian phylogeny of *Bungarus* species. This figure highlights the phylogenetic relationships between *Bungarus* species in Asia. Distinct lineages of kraits of interest have been shown in uniquely coloured boxes. Branches with superior node support (BPP ≥ 0.95) are shown in thick black lines, and the BPP values for ND4 and cyt *b* markers are indicated for the clades of interest. Branch lengths in the tree are scaled by the number of nucleotide substitutions per site. Photographs depicting the lateral view of *B. fasciatus, B. sindanus*, *B. romulusi* and *B. caeruleus* have also been shown.

**Figure 3 toxins-13-00069-f003:**
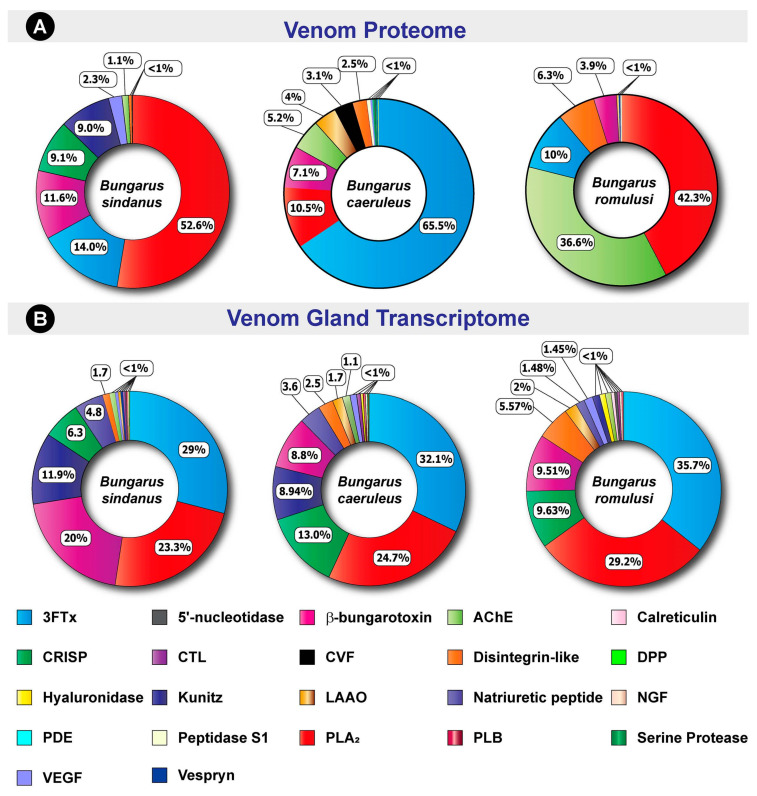
The comparative venom proteomic and venom gland transcriptomic profiles of *B. sindanus*, *B. caeruleus* and *B. romulusi*. Here, doughnut charts depict the relative abundance of various toxins in (**A**) the venom proteome and (**B**) the venom gland transcriptomes of the three cryptic kraits. Individual toxins are uniquely colour coded, and their relative abundances are indicated in percentages.

**Figure 4 toxins-13-00069-f004:**
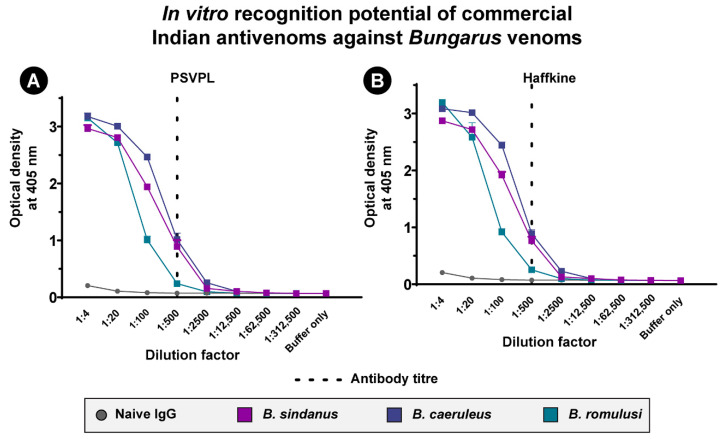
The in vitro recognition potential of commercial Indian antivenoms against *Bungarus* venoms. The venom recognition potential of naive horse Immunoglobulins G (IgG) and commercial Indian antivenoms (**A**) PSVPL and (**B**) Haffkine against the venoms of *B. sindanus*, *B. caeruleus* and *B. romulusi*, determined using end-point ELISA, is shown here. The absorbance values are plotted as mean of triplicates, and the standard deviation is shown in the form of error bars.

**Figure 5 toxins-13-00069-f005:**
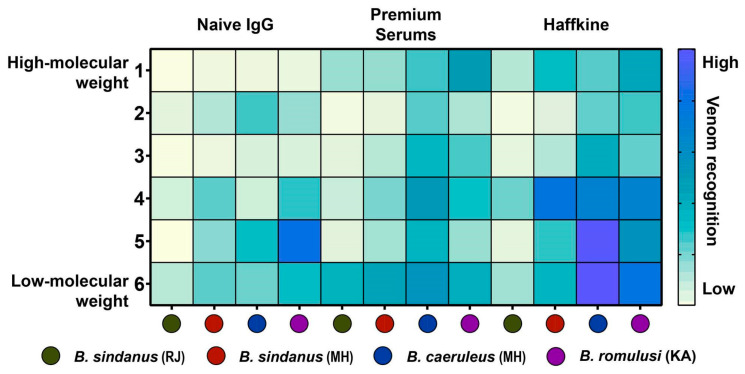
The venom recognition potential of commercial Indian antivenoms and naive horse IgGs against *Bungarus* venoms, as revealed by Western blotting experiments are depicted here. Densitometric analyses of immunoblot bands (1 to 6) are plotted as a heatmap, and a gradient of yellow to blue colouration indicates low to high venom recognition potential, respectively.

**Figure 6 toxins-13-00069-f006:**
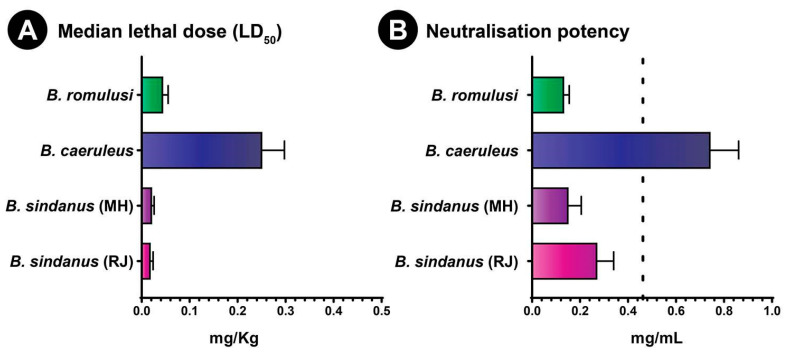
The toxicity profiles of *Bungarus* venoms and the neutralisation potencies of commercial Indian antivenoms. This figure shows (**A**) venom potencies (mg/Kg) of kraits and (**B**) the neutralisation potency (mg/mL) of PSVPL antivenom against them. Here, the error bars represent 95% confidence intervals, while the vertical dotted line in panel B indicates the marketed potency of ‘big four’ antivenoms against *B. caeruleus* (0.45 mg/mL). LD_50_ of *B. sindanus* from Rajasthan and the neutralisation potency value against it are from [15].

**Figure 7 toxins-13-00069-f007:**
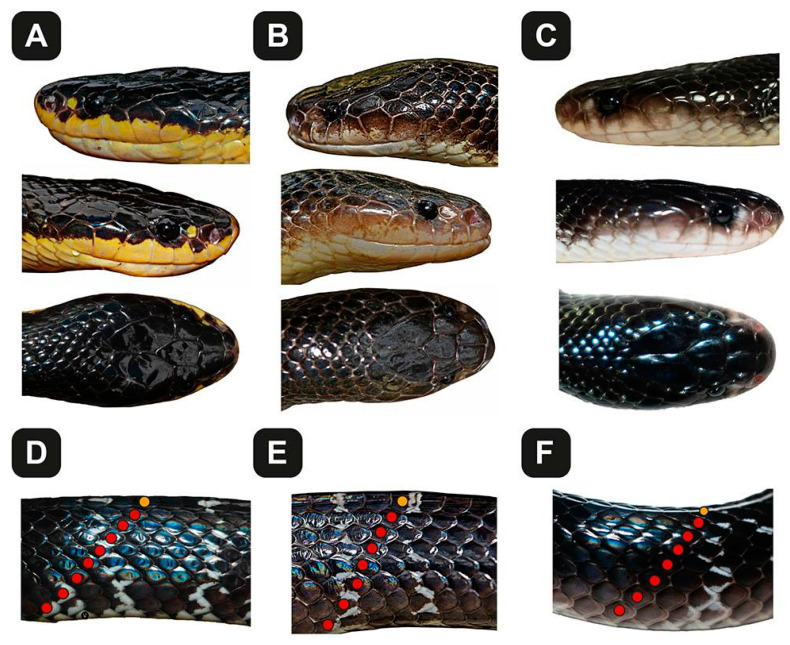
Phenotypic similarities and differences in the cryptic kraits of Southern and Western India. These photographs highlight the morphological differences between (**A**) *B. sindanus*, (**B**) *B. caeruleus* and (**C**) *B. romulusi*. The images in the panels (**D**–**F**) show the lateral view of the mid-body, highlighting the edges of the ventral scales, the adjacent dorsal scale rows (7 or 8 red dots) and the uppermost dorsal scale row (yellow dot). The corresponding scale rows on the other side are not visible, and the total mid-body dorsal scale count (i.e., 17 in *B. sindanus* and 15 in *B. caeruleus* and *B. romulusi*) is by inference. Full-body images of *B. romulusi* have also been deposited in MorphoBank (#3897).

## Data Availability

The *B. romulusi* image data presented in this study is openly available in Morphbank (#3897). The DNA sequence data generated is accessible through NCBI Genbank database. The raw proteomics data can be found at PRIDE Database (Accession No: PXD023127). The transcriptomics data presented in this study can be openly accessed via Sequence Read Archive (SRA) at NCBI (Bioproject: PRJNA681550; SRA: SUB8677374, SUB8648213 and SUB8655883).

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
