# Peer review of "A Wolf in Another Wolf’s Clothing: Post-Genomic Regulation Dictates Venom Profiles of Medically-Important Cryptic Kraits in India"

_toxins, 2021, doi:10.3390/toxins13010069_

Round 1
Reviewer 1 Report
Snakebite is an important disease of the rural tropical poor that kills approx. 138,000 people annually, with estimations of 50,000 annual deaths in India alone, and many times more victims survive with lifelong disabilities.
The authors have carried out very important research that demonstrates why the burden of mortality and morbidity from snakebite persists despite availability of antivenom. It demonstrates the value, and need for further comprehensive understanding of species identities, venom profiles and to determine regional variation among snake species. This is important for immediate development of regional antivenoms and in the future for designing next generation broad spectrum antivenoms capable of neutralising the clinical manifestations of envenoming’s by sympatric species.
It was a pleasure to read this very interesting study that was carried out and written to a very high standard.
Suggestions:
Line 30: Suggest changing "Bite" to "Bites"
Line 73-74: For readers who are not immediately familiar with the locations, I suggest adding the locations for Rajasthan, Maharashtra and Karnataka to the map in Figure 1A. Also name the locations for the circles depicting the isolated records.
Line 86: Suggest adding West Bengal to the map in Figure 1A.
Reviewer 2 Report
The paper is thorough and multidisciplinary, covering phylogeny, transcriptomics, proteomics, and toxicology. The species in question are medically significant. The authors have elucidated the phylogentic relationships of the Bungarus clade on the Indian subcontinent, and made a comparative analysis of their venom compositions. I am happy to approve for publication, although I would recommend the following minor textual changes;
Line 5,6 – remove inverted comma’s from “phenotypically similar”
Line 23 – change “alarming” to “poor”
Line 60 – change “Thus we emphasise” to “Therefore we highlight”
Line 157-158 – change “a heterodimeric toxin with PLA2 and Kunitz activities” to “a presynaptic neurotoxin which is a heterodimer of a PLA2 and a Kunitz peptide”
Line 439 – change “colossal” to “large”
Line 445-446 – change “Thus, there is a pressing need to produce regionally effective antivenoms in India to safeguard the lives of hundreds of thousands of snakebite victims.”, to “Given the high population densities in this region, there is a need for a more regionally specific antivenom”.
Reviewer 3 Report
A wolf in another wolf’s clothing: Post-genomic regulation dictates venom profiles of medically-important cryptic kraits in India
In the present study, the authors conducted phylogenetic and comparative venomics investigations of kraits in Southern and Western India to understand the distribution of genetically-distinct lineages of kraits, compositional differences in their venoms, and the consequent impact of venom variation on the (pre)clinical effectiveness of antivenom therapy. In my opinion, the study is still interesting and innovative, including was well delineated. However, I have some comments:
Comment (1): In my opinion, title is good but it should be changed to be more attractive.
Comment (2): Abstract. The background topic is poor. It is necessary include at least one more clear objective for study. Unfortunately, I cannot catch the aim of work. In addition, a more concluded sentence is required.
Comment (3): Introduction. There is a brief review of existing knowledge and relevance of study but it is missing more information. I suggest to the authors to include some reports about the number of victims of these snakes in India.
- General comment: Proofreading is strongly recommended to ensure that the text is easily understandable.
Comment (4): Results. Very good written results. A lot of information are included.
Figure 1A. I recommend to the authors to add clear photos of the snakes. Actually, I cannot see what they are included.
Figure 1B. Could you please explain to me what protein families are present in the venom? Add a little description in the figure legend.
Comment (5): Discussion. Great information in this section but need a proofreading to improve it well.
Comment (6): Conclusion. The authors concluded their works at the end systematically.
Comment (7): Materials and Methods. It is clear and the details are sufficient to understand the strategies adopted for the development of the study.
Comment (8): Figures and legends. Legends are adequate and figures are necessary to understand the results obtained.
